# FARKAS LAYERS: DON'T SHIFT THE DATA, FIX THE GEOMETRY

## ABSTRACT

Successfully training deep neural networks often requires either batch normalization, appropriate weight initialization, both of which come with their own challenges. We propose an alternative, geometrically motivated method for training. Using elementary results from linear programming, we introduce Farkas layers: a method that ensures at least one neuron is active at a given layer. Focusing on residual networks with ReLU activation, we empirically demonstrate a significant improvement in training capacity in the absence of batch normalization or methods of initialization across a broad range of network sizes on benchmark datasets.

## 1 INTRODUCTION

The training process of deep neural networks has gone through significant shifts in recent years, primarily due to revolutionary network architectures, such as Residual neural networks (He et al., 2016) and *normalization* techniques, initially proposed as batch normalization (Ioffe & Szegedy, 2015). The former is the backbone for current "state-of-the-art" results for image-based tasks such as classification. Normalization can be found in a variety of flavors and applications; (Ba et al., 2016; Ulyanov et al., 2016; Wu & He, 2018; Vaswani et al., 2017; Zhu et al., 2017). For reasons that are not fully understood, normalization gives rise to many desirable traits in network training, e.g. a fast convergence rate, but also comes with a cost by increasing vulnerability to adversarial attacks (Galloway et al., 2019). Apart from normalization, network weight *initialization* has been a driving force of improved performance. Throughout the years, various initialization schemes have been proposed, in particular Xavier initialization (Glorot & Bengio, 2010), FixUp (Zhang et al., 2019), and a recent asymmetric initialization (Lu et al., 2019). These approaches are often connected to probabilistic notions, or balancing learning rates while training.

Modern network architectures catered for image-classification tasks incorporate various one-sided activation functions, the canonical example being the Rectified Linear Unit (ReLU) (Nair & Hinton, 2010). Other one-sided activations include the Exponential Linear Unit (ELU) (Clevert et al., 2015), Scaled ELU (SELU) (Klambauer et al., 2017), and LeakyReLU (Maas et al.). ELU and SELU were constructed with the purpose of eliminating batch normalization (BN); these activation functions minimize the internal covariate shift (ICS), which is what BN was intended to accomplish. Recent developments have shown that BN has little impact on ICS, but affects the smoothness of the loss landscape, allowing for easier (and faster) training regimes (Santurkar et al., 2018).

A geometric interpretation of batch normalization can be readily seen in the context of the *dying ReLU problem*. This phenomenon occurs when the gradient of the neuron is zero (when the neuron outputs only negative values), in which case the neuron is "dead". This effectively freezes the neuron during training. If too many neurons are dead, the network learns slowly. In fact, it is entirely possible for a network to be "born dead", where it does not allow learning at all (Lu et al., 2019). To motivate this, consider a simple binary classification problem: suppose two sets of points in $\mathbb{R}^2$ are sufficiently separated and we wish to classify them using a simple 1-layer ReLU network (with standard Cross Entropy loss). If the network is initialized such that one cloud of points is not "observed", as in Figure 1a, the network will not learn to classify those points. Batch normalization will shift the points to behave like 1b; the network is no longer dead. There is a far simpler geometric solution in $\mathbb{R}^2$: given one hyperplane, construct another to face the missing data,

as shown in Figure 1c by the green hyperplane, which is now a non-dead component of the network.

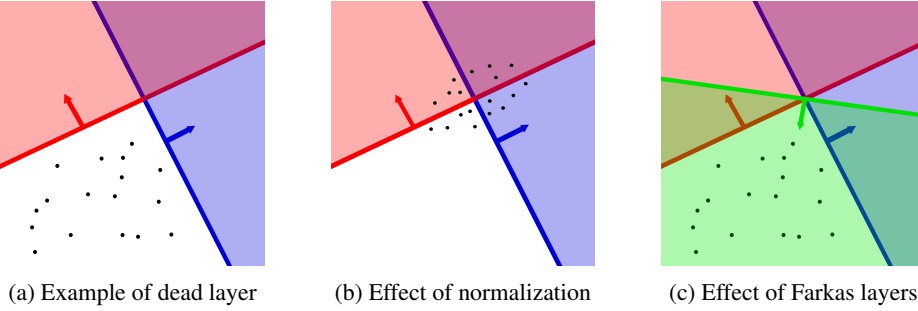

(a) Example of dead layer     (b) Effect of normalization     (c) Effect of Farkas layers

Figure 1: A motivating geometric interpretation

### OUR CONTRIBUTIONS

We present a geometrically motivated method for network training in the absence of initialization and normalization. Our novel layer structure, called Farkas layers, ensures that *at least* one neuron is active in every layer. Using off-the-shelf networks, Farkas layers can recover a significant part of the explanatory power of DNNs when batch normalization is removed, sometimes up to 10%, as demonstrated by empirical results on benchmark image classification tasks, such as CIFAR10 and CIFAR100. When used in conjunction with batch normalization on ImageNet-1k networks, we empirically show an approximate 20% improvement on the first epoch over only using batch normalization. Finally, we claim that input data normalization is a critical component for using large learning rates in FixUp, which is not the case for Farkas layer-based networks. All in all, this work provides a new research direction for architecture design beyond initialization methods, normalization layers, or new activation functions.

## 2 BACKGROUND

### 2.1 NOTATION

We define neural networks as iterative compositions of activation functions with linear functions. More specifically, given an input $x^{(k)} \in \mathbb{R}^n$ to the $k$-th layer, the output $x^{(k+1)}$ is typically written

$$x^{(k+1)} = \varphi(W^{(k)}x^{(k)} + b^{(k)})$$

where $W^{(k)} \in \mathbb{R}^{m \times n}$ is a weight matrix, the rows are written as $w_i^\mathsf{T}$ $(i = 1, \ldots, m)$, $b^{(k)} \in \mathbb{R}^m$ is the layer's bias, and $\varphi$ is the activation function of the layer. For the remainder of our analysis, we consider the ReLU activation function, written

$$\varphi(x) = \max\{x, 0\},$$

which acts component-wise in the case of vector arguments.

### 2.2 RELATED WORK

Previous work on the dying ReLU problem, or vanishing gradient problem, in the case of using sigmoid-like activation functions, has heavily revolved around novel weight initializations. To address the vanishing gradient problem, (Glorot & Bengio, 2010) proposed an initialization that is still often used with ReLU activations (Shang et al., 2017). (He et al., 2015) proposed a scaled version of a normal distribution, which improved training for pure convolutional neural networks. More recently, (Lu et al., 2019) studied the dying ReLU problem from a probabilistic perspective and addressed networks that are "born dead", i.e. not capable of learning. Their analysis is based on the reasonable assumption that all weights and biases are initialized with a non-zero probability of being dead:

$$\mathbb{P}\left(w_i^\mathsf{T} x + b_i < 0\right) \geq p > 0 \quad (i = 1, \ldots, m),$$

where $m$ is the number of neurons in a given layer, and $p$ is a fixed positive constant. They show that, as the number of layers approaches infinity, the probability of having a network be born dead approaches one. They also find that initializing weights with a symmetric distribution is more likely to cause a network to be born dead and thus propose initializing weights using asymmetric distributions to mitigate this problem. Their analysis is relevant in the case that the network is not very wide, which is distinct from the case we study in this paper.

Normalization has been one of the driving forces of recent state-of-the-art results; where the general idea is to manipulate the neuron activation with various statistics, such as subtracting the mean and dividing by the variance. Popular normalization techniques include batch normalization (Ioffe & Szegedy, 2015), Layer normalization Ba et al. (2016), Instance normalization (Ulyanov et al., 2016) and Group normalization (Wu & He, 2018). However, there is a desire to eliminate normalization from training. A recent work that does this is FixUp (Zhang et al., 2019), a novel initialization technique that works by scaling the weights as a function of the network structure, allowing the network to take large learning rate steps during training. Reportedly, FixUp provides the same level of accuracy as that of a BN-enhanced network, and scales to networks that train on ImageNet-1k. As stated in FixUp, the inherent structure of residual networks already prevents some level of gradient vanishing, since the variance of the layer output grows with depth. In the case of positively-homogeneous blocks (e.g. no bias in a Linear or Convolutional layer), this type of variance increase can lead to gradient explosion (Hanin & Rolnick, 2018), which makes training more difficult.

## 3 FARKAS LAYERS

### 3.1 AUGMENTING RELU ACTIVATED LAYERS

We present a novel approach to understanding how weight matrices and bias vectors contribute to neural network learning, called Farkas layers. As the name suggests, our method is influenced by Farkas' lemma. In particular, we use the following representation of Farkas' lemma; the proof is an exercise in linear programming and is left in the appendix.

**Lemma 3.1** (Representation of Farkas' lemma). *Let $W \in \mathbb{R}^{m \times n}, b \in \mathbb{R}^m$ with $x \in \mathbb{R}^n$. Then the set $\{x \mid Wx + b \leq 0\}$ is empty if and only if there exists a $\lambda \in \mathbb{R}^m$ such that $\lambda_i \geq 0$ (for $i = 1, \ldots, m$), with $W^\mathsf{T}\lambda = 0$ and $\lambda^\mathsf{T} b > 0$.*

We present the construction of $W$ and $b$, and prove using Lagrangian duality (Boyd & Vandenberghe, 2004) that such weights and bias vectors will satisfy Farkas' lemma, resulting in at least one positive component. In essence, this solves the dying ReLU problem while training.

**Theorem 3.2.** *Let $L$ be a layer of a neural network with ReLU activation function, with weights $W \in \mathbb{R}^{m \times n}$ (with rows $w_i^\mathsf{T}$) and bias vector $b \in \mathbb{R}^m$, and let $\lambda \in \mathbb{R}^m$, with $\lambda_i \geq 0$ (with at least one $\lambda_i > 0$, without loss of generality $\lambda_m > 0$). Further, suppose $W$ has the property that*

$$w_m = \frac{-1}{\lambda_m} \sum_{j=1}^{m-1} \lambda_j w_j$$

*and that $b_m > \frac{-1}{\lambda_m} \sum_{j=1}^{m-1} \lambda_j b_j$. Then this layer has a non-zero gradient.*

*Proof.* Let $W \in \mathbb{R}^{m \times n}$ (with rows $w_i^\mathsf{T}$) and bias $b \in \mathbb{R}^m$, with the aforementioned assumptions, and arbitrary input $x$. The output of this layer is $\varphi(Wx + b)$, and has a non-zero gradient if and only if there exists at least one component such that

$$p^* := \max_i w_i^\mathsf{T} x + b_i > 0. \tag{1}$$

The left-hand side of (1) (i.e. omitting the strict inequality constraint) can be written as the following minimization problem:

$$p^* := \min_{x,s} s \quad \text{subject to} \quad w_i^\mathsf{T} x + b_i \leq s \ (i = 1, \ldots, m),$$

or equivalently,

$$p^* = \min_{x,s} s \quad \text{subject to} \quad Wx + b \leq s\mathbf{1},$$

where $\mathbf{1}$ is the vector of all ones. The Lagrangian for this problem is

$$\mathcal{L}(x, s, \lambda) = s + \lambda^\mathsf{T}(Wx + b - s\mathbf{1})$$

where $\lambda \geq 0$. We compute the dual problem,

$$
\begin{aligned}
d(\lambda) &= \inf_{x,s} \mathcal{L}(x, s, \lambda) \\
&= \inf_{x,s} s(1 - \lambda^\mathsf{T}\mathbf{1}) + \lambda^\mathsf{T}b + (W^\mathsf{T}\lambda)^\mathsf{T}x \\
&= \begin{cases} \lambda^\mathsf{T}b & \text{if } \lambda^\mathsf{T}\mathbf{1} = 1, W^\mathsf{T}\lambda = 0, \ (\lambda \geq 0) \\ -\infty & \text{otherwise.} \end{cases}
\end{aligned}
$$

The conditions on $W$ are met by assumption, and there are many ways to ensure that $\lambda$ is on the simplex, and thus has at least one strictly positive component. Both the primal and dual problems are linear optimization problems: the objective and constraints are linear. By weak duality, $p^* \geq \sup_\lambda d(\lambda)$. Thus to ensure that $p^* > 0$ (as in (1)), it suffices to show that $\lambda^\mathsf{T}b > 0$. This is guaranteed by the assumptions on the bias vectors, and so we are done. $\qquad\square$

It is highly unlikely that a given layer in a deep neural network has all dead neurons. However, we believe that guaranteeing the explicit activity of one neuron is beneficial in that it improves training. Figure 1 motivates the use of Farkas layers as a potential substitute for (BN) in the context of learning: the arrows of the corresponding hyperplane indicate the "non-zero" part of the ReLU. In the case of 1a, we see that all data points end up on the "zero" side, which would typically lead to no learning. Heuristically, the effect of BN mimics Figure 1b, where the data points are centered and scaled by a diagonal matrix (which geometrically acts like an ellipse) allowing for learning to occur. Farkas layers will instead append another hyperplane, that will be guaranteed to see all the points (in a "third" dimension).

### 3.2 EXTENSION TO GENERAL ONE-SIDED ACTIVATIONS

The use of Farkas layers can be extended to other one-sided activation functions, such as a smooth-ReLU (i.e. replacing the kink with a polynomial), ELU, and several others. In all these cases, one can impose a scalar cutoff value $c$ such that "beyond" this cutoff, we have very small (or zero) gradients. This is shown in the following corollary, which has the same proof as Theorem 3.2.

**Corollary 3.3.** *Let $L$ be a layer of a neural network with an arbitrary one-sided activation function $\psi$ with cutoff $c \in \mathbb{R}$, i.e. for all $x < c$, we have $\psi(x) \simeq 0$ (or exactly zero). Denote the weights by $W \in \mathbb{R}^{m \times n}$ (with rows $w_i$) and bias vector $b \in \mathbb{R}^m$, and let $\lambda \in \mathbb{R}^m$, with $\lambda_i \geq 0$, with at least one positive component. To ensure that $\psi(Wx + b)$ has at least one component greater than $c$, we require that*

$$w_m = \frac{-1}{\lambda_m} \sum_{j=1}^{m-1} \lambda_j w_j$$

*and that $b_m > c - \frac{1}{\lambda_m} \sum_{j=1}^{m-1} \lambda_j b_j$.*

### 3.3 ALGORITHM AND LIMITATIONS

We denote a Farkas layer by $\mathcal{F} := \mathcal{F}(\cdot; W, b) : \mathcal{D}_{\text{in}} \to \mathcal{D}_{\text{out}}$, where the weights and biases satisfy Theorem 3.2. The construction of $W$ and $b$ imposes certain restrictions on the existence of Farkas layers, namely we require that $\dim(\mathcal{D}_{\text{out}}) \geq 2$. Finally, some network architectures claim to perform better without bias vectors present; we require bias vectors by construction. We focus on replacing Convolutional and Linear layers in (deep) neural networks, with efficient implementations in PyTorch. The general method of incorporating Farkas layers is provided in Algorithm 1. To minimize computational cost, we consider $\lambda$ to be the evenly spaced simplex element i.e. $\lambda_i = \dim(\mathcal{D}_{\text{out}})^{-1}$ (for $i = 1, \ldots, \dim(\mathcal{D}_{\text{out}})$), but for smaller problems (such as the

motivating 1-layer example) $\lambda$ can be learnable. Farkas layers can be made into Farkas blocks in conjunction with residual networks as well, outlined in Algorithm 2, which we leave in the Appendix. In the case of multiple convolutional layers in a given residual block, the extension is natural.

In both Algorithm 1 and 2, we consider two choices for the aggregating function (AggFunc): either the sum or the mean of the previous outputs and biases. Indeed, let $\dim(\mathcal{D}_{\text{out}}) = m$, and thus $\lambda_j = m^{-1}$. Then we have the following two aggregation methods:

$$w_m = -\sum_{j=1}^{m-1} w_j \quad \text{(sum)}, \qquad w_m = \frac{-1}{m-1} \sum_{j=1}^{m-1} w_j \quad \text{(mean)},$$

and similarly for the bias vectors. Theorem 3.2 is constructed using "sum", but the result holds in the case of using "mean" since ReLUs are 1-positive homogenous. Furthermore, by using "mean", we induce stability with respect to the $\ell_\infty$-induced matrix norm for a matrix $A \in \mathbb{R}^{m \times n}$, defined as

$$\|A\|_\infty = \max_{1 \le j \le m} \|a_j^\mathsf{T}\|_1,$$

where $a_j^\mathsf{T}$ are the rows of $A$. Let $\tilde{A} \in \mathbb{R}^{(m-1) \times n}$ and $\boldsymbol{a}^\mathsf{T}$ be the mean of the rows of $\tilde{A}$. Define $A := (\tilde{A}; \boldsymbol{a}^\mathsf{T}) \in \mathbb{R}^{m \times n}$, then

$$\|A\|_\infty = \max\left\{\|\tilde{A}\|_\infty, \|\boldsymbol{a}^\mathsf{T}\|_1\right\} \le \max\left\{\|\tilde{A}\|_\infty, (m-1)^{-1} \sum_{j=1}^{m-1} \|a_j^\mathsf{T}\|_1\right\} \le \|\tilde{A}\|_\infty.$$

Thus, in the context of network weights, the burden of stability lies on the trainable weights and not the aggregated component. A similar analysis can be done for the bias vectors. Thus, for networks that train on ImageNet-1k, or to potentially use larger learning rates, we believe that using "mean" as the aggregation function is necessary.

---

**Algorithm 1** FarkasLayer with ReLU activation; $\mathcal{F}(\cdot; W, b)$ with AggFunc = "sum" or "mean"

---

Initialize $W \in \mathbb{R}^{(m-1) \times n}$ and $b \in \mathbb{R}^m$
Forward pass: $x$
$y = Wx$                           ▷ e.g. Linear and/or Conv with no bias
Compute the Aggregated Components:
$\boldsymbol{y} := -\text{AggFunc}(y)$
$\boldsymbol{b} := \max\{-\text{AggFunc}(b[0:-1]), b_m\}$
$y \leftarrow \text{concatenate}(y, \boldsymbol{y})$
$b \leftarrow \text{concatenate}(b[0:-1], \boldsymbol{b})$
$y \leftarrow y + b$
$y \leftarrow \text{ReLU}(y)$
Optional: apply batch normalization to $y$
Return $y$

---

## 4    EXPERIMENTS — IMAGE CLASSIFICATION

Image classification is a standard benchmark for measuring new advancements in deep learning. Our goal is to show how training with Farkas layers, henceforth referred to as FLs, impacts learning in these settings relative to other training techniques such as normalization and initialization (e.g. FixUp).

We consider CIFAR10, CIFAR100, and ImageNet-1k datasets for the purposes of testing FL-based residual networks on image classification tasks. For CIFAR10, we consider 18, 34, 50, and 101 layer residual networks. The latter two of these use a BottleNeck block structure. On CIFAR100, we only consider 34, 50, and 101 layers. All models are trained using the same SGD schedule for 200 epochs. We make the distinction between a "large" learning rate, meaning 0.1, and a "small" learning rate, which is 0.01. We use standard data augmentation (RandomCrop and RandomHorizontalFlip), but do *not* normalize the inputs with respect to the mean and standard

deviation of the dataset in the case of CIFAR10/CIFAR100. The weights are left at the default PyTorch initialization. Additionally, we use cutout to improve generalization (DeVries & Taylor, 2017), and use weight-decay. For ImageNet-1k, we use modified code from the DAWNBench competition (Coleman et al., 2018; Shaw et al.). In this set of experiments, we augment the data via RandomCrop, RandomHorizontalFlip, and normalize the input data. We only consider the case of networks using BN and see how FLs can improve performance, and train with 30 epochs.

Henceforth, we call a residual network a FarkasNet if is it comprised of only FLs, and we always use the "sum" aggregation function unless otherwise specified. We make the distinction of networks that use or do not use BN in the tables. When omitting BN, we use the small learning rate. For CIFAR10 and CIFAR100, the results presented are averaged across three runs; if one of the runs failed[1] to train, we omit it from the average but place an asterisk. Full training curves are in the Appendix.

INTERPRETING DEPENDENCE OF BATCH NORMALIZATION

Table 1: Percent misclassification comparison for CIFAR10 (lower is better)

|  | 18 Layers | 34 Layers | 50 Layers | 101 Layers |
|---|---|---|---|---|
| ResNet (with BN, large LR) | 7.01 | 6.79 | 5.04 | **4.95** |
| FarkasNet (with BN, large LR) | **6.87** | **6.50** | **5.03** | 4.98 |
| ResNet (no BN, small LR) | 12.51 | 9.90 | 24.53 | 25.25 |
| FarkasNet (no BN, small LR) | **10.83** | **9.45** | **19.84** | **15.38** |

Table 2: Percent misclassification comparison for CIFAR100 (lower is better)

|  | 34 Layers | 50 Layers | 101 Layers |
|---|---|---|---|
| ResNet (with BN) | 30.95 | **23.43** | **22.90** |
| FarkasNet (with BN) | **30.34** | 23.88 | 23.22 |
| ResNet (no BN) | 40.10 | 55.05* | 54.02 |
| FarkasNet (no BN) | **34.70** | **43.93** | **42.94** |

Tables 1 and 2 both show a strong disparity in the learning capacity of a DNN when batch normalization is removed, which is unsurprising. On CIFAR10, we primarily observe similar, if not better, test errors when batch normalization is used. Without normalization, our test error is always better; the same can be said for the CIFAR100 dataset. Thus, while FarkasResNets also diminish in quality, we note that adding a guaranteed undead neuron to all layers heavily impacts learning in the case of a non-normalized network. This is not at all to say that training a deep neural network is only possible using FLs, or that we achieve state-of-the-art performance; however, our implementation demonstrates the strong dependency of getting low test accuracy with normalization, and is geometrically motivated.

IMPROVEMENT AT FIRST EPOCH FOR IMAGENET-1K

The final table shows similar behaviour on ImageNet-1k, where FLs[2] neither dramatically improve nor inhibit the learning capacity towards the end of training except in the case of 50-layers, where it performs significantly better. Figure 2 shows that FarkasNets have a dramatic advantage over standard ResNets at the start of training. Evidently, this advantage gradually diminishes; nonetheless, its presence alludes to a bigger problem in network initialization. We notice a roughly 20% improvement on the first epoch, simply due to the use of FLs.

---

[1]Failed implies that the network did not make any progress in the first five epochs.
[2]For ImageNet-1k, we use "mean" aggregation function.

Table 3: Top1/Top5 percent misclassification comparison for ImageNet-1k (lower is better)

|  | 50 Layers | 101 Layers | 152 Layers |
|---|---|---|---|
| ResNet (with BN) | 25.64/7.68 | **23.23/6.49** | 22.26/5.95 |
| FarkasNet (with BN) | **23.70/6.65** | 23.74/6.67 | **22.17/5.86** |

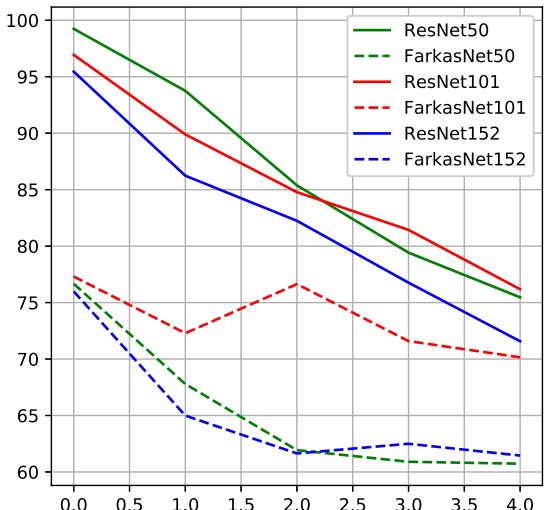

Figure 2: Illustration of improved performance at starting epochs with Farkas layers

IMPACT OF DATA NORMALIZATION AND BEST PRACTICE COMPARISON

We briefly compare against other best practice learning regimes on CIFAR10. Table 4 highlights the various differences between methods. At the core, we compare FarkasNets to FixUp, a recently proposed initialization scheme that allows the use of larger learning rates in the absence of batch normalization, and thus faster convergence. We consider a 34-layer ResNet with BasicBlock structure using the official implementation of FixUp[3] for the network architecture. We compare using a 34-layer FarkasNet, where the last layer in a block is initialized to zero (as in FixUp, which is motivated by several other works (Srivastava et al., 2015; Hardt & Ma, 2016; Goyal et al., 2017)) but use standard initialization otherwise, as well as "mean" AggregationFunction. We maintain the setup of the previous experiments.

Data normalization is a standard "trick" for training deep neural networks, and acts like an initial batch normalization step, where the incoming data is scaled to have zero mean and unit variance. In the case of CIFAR10 and CIFAR100, we strove to study the problem of training without any attempts at normalization, which is why we have omitted this from our training methodology. In the absence of data normalization, we observed that FixUp requires a small learning rate for training. Thus, while (Zhang et al., 2019) claims to present a theory for addressing training in the absence of batch normalization, we find that it still heavily hinges on normalization principles, namely on the first layer pass. This observation is not to diminish the benefits of FixUp, as faster convergence is observed (see Figure 3), but the learning rate does need adjustment. On the other hand, our 34-layer FarkasNet was able to use a medium learning rate of 0.05, allowing for improved performance relative to Table 1.

---

[3]From one of the authors' Github pages

Table 4: Best practice comparison on CIFAR10 (lower test error is better)

|  | Batch Normalization | LR capacity | Test Error |
|---|---|---|---|
| ResNet34 | ✓ | Large | 6.77 |
| ResNet34 | ✗ | Small | 9.18 |
| FixUp34 | ✗ | Small | 7.49 |
| FarkasNet34 | ✗ | Medium | 7.12 |

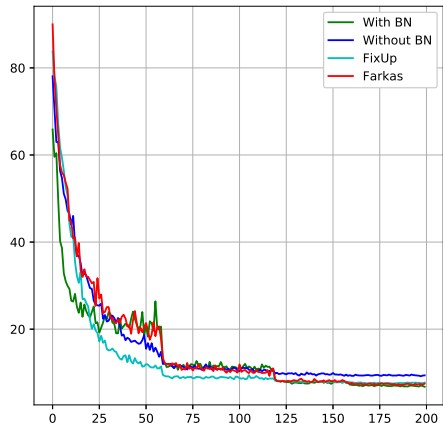

Figure 3: Training curves for best practice comparison on CIFAR10

Finally, we also address the notion that training a *very* deep neural network is challenging using maximal learning rate without normalization, which has been experimentally observed in FixUp and Layer-Sequential Unit-Variance (LSUV) orthogonal initialization (Mishkin & Matas, 2015). Despite the shortcomings of previous work, by incorporating the "mean" aggregation function and default initialization, we are able to successfully train a 101-layer FarkasNet with large learning rate on CIFAR10. We present the averaged training curve in Figure 5, left in the Appendix, where we repeated the experiment three times. We remark that using default initialization alone results in complete failure to train.

## 5 CONCLUSION

To date, successful training of a deep neural networks requires either some form of normalization, weight initialization, and/or choice of activation function, all of which come with their own challenges. In this work, we provide a fourth option, Farkas layers, which can be used alone or in conjunction with the aforementioned techniques. The Farkas layer is based on the interaction of the geometry of the weights with the data: by simply adding one linearly dependent row to the weight matrix, we ensure that no neurons are dead. Using our method, we have shown that training with larger learning rates is possible even in the absence of batch normalization, and with default initialization. In this work, we only touched on image classification, but Farkas layers could be used in many deep learning applications, which is left for future work.

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

# A  APPENDIX

## A.1  ELEMENTARY LINEAR PROGRAMMING

**Proposition A.1** (Farkas' Lemma)**.** *Let $W \in \mathbb{R}^{m \times n}$ and $b \in \mathbb{R}^m$.*

$$\{x \mid Wx + b = 0, x \geq 0\} \neq \emptyset \iff \nexists \lambda \text{ s.t. } \lambda^\mathsf{T} W \geq 0, \ \lambda^\mathsf{T} b > 0.$$

**Corollary A.2.** *Let $W \in \mathbb{R}^{m \times n}$ and $b \in \mathbb{R}^m$. Then*

$$\{x \mid Wx + b \leq 0\} = \emptyset \iff \exists \lambda \text{ s.t } \lambda^\mathsf{T} W = 0, \ \lambda^\mathsf{T} b > 0 \quad (\lambda \geq 0).$$

*Proof.* Suppose by contradiction there exists an $x$ such that $Wx + b \leq 0$. Let $x^+ := \max\{x, 0\}$ and $x^- := \max\{-x, 0\}$, and so $x = x^+ - x^-$. Then:

$$\exists x \text{ s.t } Wx + b \leq 0$$
$$\iff \exists x^+, x^- \text{ s.t } W(x^+ - x^-) + b \leq 0$$
$$\iff \exists x^+, x^-, \lambda \text{ s.t } W(x^+ - x^-) + \lambda + b = 0 \quad (\lambda \geq 0)$$
$$\iff \bar{x} := (x^+, x^-, \lambda)^\mathsf{T} \geq 0, \ \bar{W} := [W \mid -W \mid I] \text{ s.t } \bar{W}\bar{x} + b = 0,$$

which follows from Farkas' Lemma. $\square$

## A.2  ALGORITHM FOR RESIDUAL FARKAS LAYER

---
**Algorithm 2** Residual Farkas layer with ReLU activation

---
Initialize $W^{(1)}, W^{(2)}, b^{(1)}, b^{(2)}$
Forward pass: $x$
$y = \mathcal{F}(x; W^{(1)}, b^{(1)})$
$y \leftarrow W^{(2)} y$  ▷ e.g. Linear and/or Conv with no bias
Compute aggregated components:
$\boldsymbol{y} := -\text{AggFunc}(x + y)$  ▷ residual component
$\boldsymbol{b} := \max\{-\text{AggFunc}(b^{(2)}[0 : -1]), b_m^{(2)}\}$
$y \leftarrow \text{concatenate}(y, \boldsymbol{y})$
$b \leftarrow \text{concatenate}(b^{(2)}[0 : -1], \boldsymbol{b})$
$y \leftarrow y + b$
$y \leftarrow \text{ReLU}(y)$
Optional: apply batch normalization to $y$
Return $y$

---

## A.3 TRAINING CURVES FOR CIFAR10

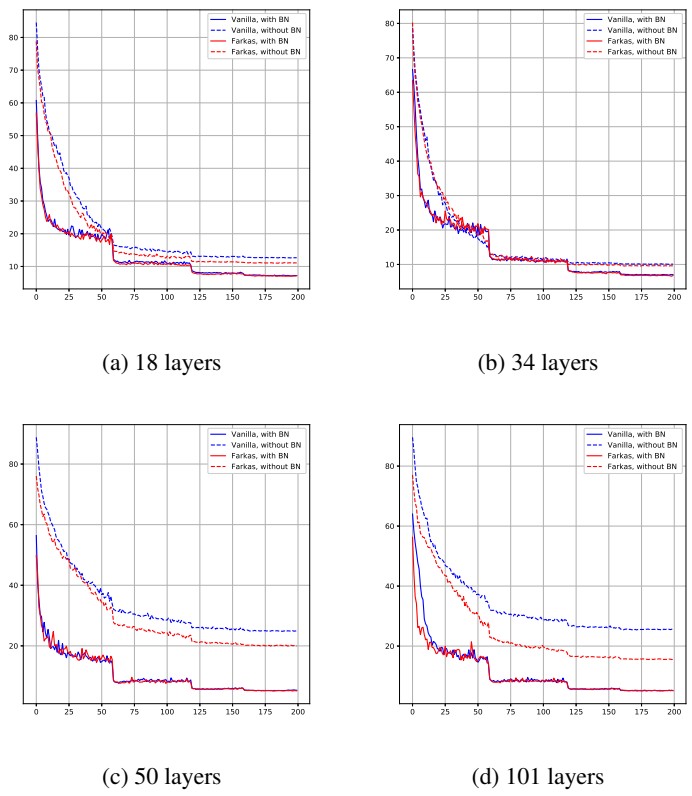

(a) 18 layers

(b) 34 layers

(c) 50 layers

(d) 101 layers

Figure 4: Average training curves for CIFAR10; "Vanilla" refers to standard ResNet. Dotted lines omit batch normalization, solid lines incorporate it.

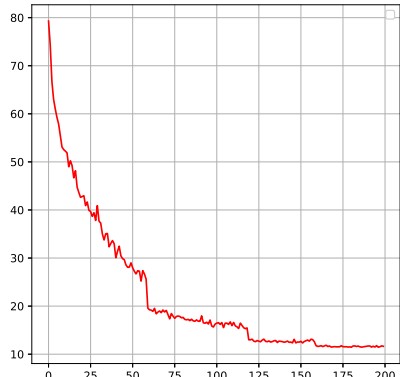

Figure 5: Average training curve for 101-layer FarkasNet, with "mean" aggregation function, maximal learning rate, and without batch normalization.

## A.4    TRAINING CURVES FOR CIFAR100

For 50-layers, one of the networks (standard ResNet without BN) failed to train, and we omit it from the average.

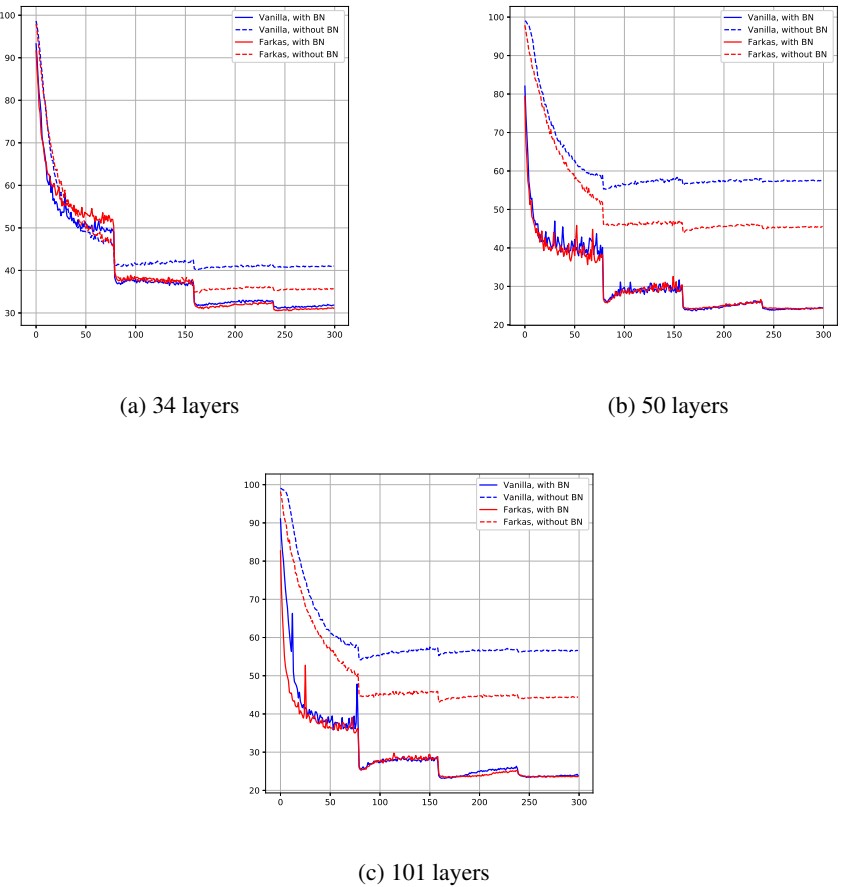

(a) 34 layers

(b) 50 layers

(c) 101 layers

Figure 6: Average training curves for CIFAR100; "Vanilla" refers to standard ResNet

## A.5    TRAINING CURVES FOR IMAGENET-1K

Here, dotted line means Farkas layer-based network.

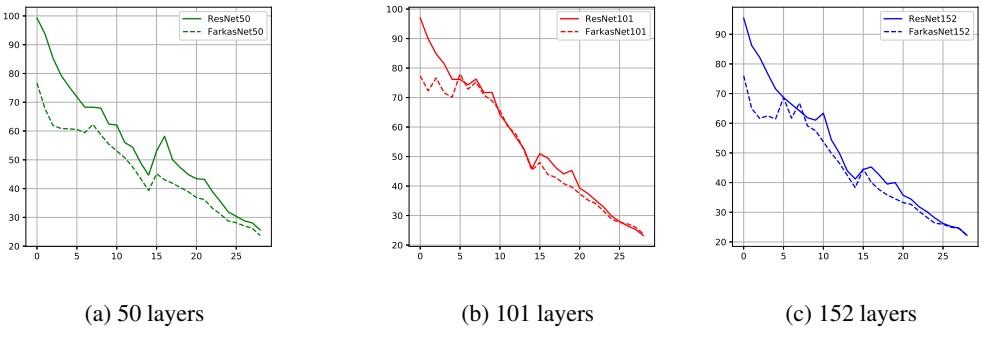

(a) 50 layers

(b) 101 layers

(c) 152 layers

Figure 7: Training curves for ImageNet

