# OpenReview forum: "Farkas layers: don't shift the data, fix the geometry"
_ICLR.cc/2020/Conference — Reject_

### Official Review · AnonReviewer2 · 2019-10-21
**Official Blind Review #2**

**Rating:** 1

**Review:**

The paper introduces a new type of layer, Farkas layers, that are designed to ameliorate the "dying ReLU problem". The idea of repurposing Farkas' lemma from linear programming to a component in NNs is very nice. However, the paper doesn't have convincing baselines and doesn't dig deep enough into what the Farkas layer is actually doing.

Comments:

1. Start of section 2.2: “Previous work on the dying ReLU problem, or vanishing gradient problem, in the case of using sigmoid-like activation functions, has heavily revolved around novel weight initializations”. Dying ReLUs and vanishing gradients are different problems. In particular, it doesn’t make sense to talk about the dying ReLU problem for sigmoid-like activation functions.


2. Batchnorm (BN) consists in two operations: shifting and scaling. Both are relevant to vanishing gradients. However, only shifting (that is, subtracting the mean and resetting to a learned value) is relevant to dying ReLUs because rescaling the input to a ReLU by a positive number doesn’t affect whether the ReLU is subsequently ON or OFF.


3. Given point #2, a natural baseline to compare the Farkas layer against is a “pared-down BN”, which shifts but does not rescale.


4. Similarly, when combining the Farkas operation with BN, it might be worth considering keeping BN’s rescaling but dropping the shift -- since Farkas is analogous to the shift operation in BN.


5. I claimed above that Farkas is analogous to the shift in BN, but haven’t thought about it deeply. Do you agree? Any comments on how they differ and why?


6. Our contributions, p2: “We empirically show an approximate 20% improvement on the first epoch over only using batch normalization.” I’m not sure what to make of this; improvements on the first epoch are only useful if they lead to overall improvements.


7. Figure 4 of “Shattered gradients”, https://arxiv.org/abs/1702.08591 looks at ReLU activations by layer, both with and without BN. It’s worth doing a similar analysis for Farkas layers. Concretely: how do Farkas layers change the pattern and frequency of activation, both at activation and during training?


8. Guaranteeing the activity of a single neuron per layer seems very weak. What is the empirical effect of Farkas on the number of live neurons? Is it really just making sure one ReLU is on, or does it do better? Is it possible to ensure more neurons are ON in each layer? The “shattered” paper above suggests BN sets close to half ReLUs as ON at initialization, and approximately controls how often ReLUs are ON or OFF during training via the shift.


9. As a broader point, the paper proposes an algorithm based on a hypothesis: that having (at least one) ReLU on helps training. It’s worth digging into the hypothesis a bit rather than just plotting training curves.


10. I would expect ResNets have much *less* of a problem with dying ReLUs than standard NNs because of the skip-connections. One would therefore expect Farkas layers to help more with standard NNs than ResNets. However, the reported results are for ResNets. What happens when there are no skip-connections?





**Experience Assessment:**

I have published one or two papers in this area.

**Review Assessment: Checking Correctness Of Derivations And Theory:**

I assessed the sensibility of the derivations and theory.

**Review Assessment: Checking Correctness Of Experiments:**

I carefully checked the experiments.

**Review Assessment: Thoroughness In Paper Reading:**

I read the paper thoroughly.

---

> ### Author Response · Authors · 2019-11-11
> **reply to reviewer #2**
>
> Thank you for the thorough critiques of our paper; we will address the comments in order.
> 1) Although they are not the same exactly, it was simply to provide context.
>
> 2) Agreed though the scaling propagates forward?
>
> 3) This would be interesting --- would you be able to provide a source?
>
> 4) Agreed!
>
> 5) It is similar in what happens. BN shifts the data, while FLs modify the weights to match the data. Inherently, they seem different to us but accomplish the same task of “let’s make this activation nice for the next layer”
>
> 7) + 8) This is a very good point; thank you for bringing this to our attention. This will definitely be incorporated in a future iteration of this paper
>
> 9) Agreed, though that was beyond the scope of the original paper.
>
> 10) There is less of a problem  with ResNets compared to standard ConvolutionalNets. We wanted to benchmark against FixUp in some cases, which is only applied to ResNets. But even on ResNets, there is still a training problem in the absence of normalization. In our experience, CNNs do not perform well even with BN, so we did not bother.

---

### Official Review · AnonReviewer1 · 2019-10-22
**Official Blind Review #1**

**Rating:** 3

**Review:**

The authors propose a new `normalization' approach called Farkas layer for improving the training of neural networks. The main idea is to augment each layer with an extra hidden unit so that at least one hidden unit in each layer will be active. This is achieved by making the extra hidden unit dependent on the rest of the units in the layer, so that it will become active if the rest are inactive, and they name it after Farkas' lemma in linear programming. This avoids the gradient becoming zero when all the units in a layer are dead.

The empirical results show that this normalization method is effective, and improves the training of deep ResNets when no batch normalization is used. The accuracies on CIFAR10 and CIFAR100 are improved with the use of Farkas layers. Unfortunately it still cannot beat or replace batch normalization. When batch normalization is used, the benefit of using this Farkas layer becomes marginal (Tables 1 and 2).

I am also not completely satisfied with the authors' explanation on why Farkas' layers work. The authors motivate the design of the layer with dead hidden units, but in the experiments they do not show if any layer actually becomes completely `dead' (or gradient becomes very small) when Farkas' layer is not used. There could be other reasons why the layer helps, other than keeping some units in a layer active.

Overall I think the idea is novel and interesting, but the improvement is not big enough to replace existing normalization methods that makes this paper slightly below the acceptance threshold in my opinion.


**Experience Assessment:**

I have published one or two papers in this area.

**Review Assessment: Checking Correctness Of Derivations And Theory:**

I assessed the sensibility of the derivations and theory.

**Review Assessment: Checking Correctness Of Experiments:**

I assessed the sensibility of the experiments.

**Review Assessment: Thoroughness In Paper Reading:**

I read the paper thoroughly.

---

> ### Author Response · Authors · 2019-11-11
> **reply to reviewer #1**
>
> Thank you for the feedback. As Reviewer #3 said, we agree that we could have done more to demonstrate the behaviour of the activation layers with or without FLs. Though, the simple geometric explanation of how the weights interact does has some empirical significance when training CIFAR10/100 models compared to no normalization whatsoever. Our goal was never to replace BN, but to provide an alternative point-of-view for how/why it works. Again, thank you for the comments.

---

### Official Review · AnonReviewer3 · 2019-10-23
**Official Blind Review #3**

**Rating:** 3

**Review:**

This paper provides a new method to deal with the zero gradient problem of ReLU (all input values are smaller than zero) when BN is removed, called Farkas layer. This Farkas layer concatenates one positive value to guarantee that, at least one neuron is active in every layer to reduce the challenge for optimization. Compared with the method without BN, Farkas layer shows better results on CIFAR-10 and CIFAR-100.

Though, I still have several concerns:
1.	The proposed Farkas layer is too simple and seems not work well. With BN, the FarkasNet does not show significant improvements than the traditional ResNet with BN on CIFAR-10, CIFAR-100 and ImageNet.
Without BN, though FarkasNet shows significant improvements than ResNet. But FarkasNet without BN cannot achieve comparable performance with FarkasNet with BN. With deeper networks, the performance further goes down, which really downgrade the rating of this paper.
In Fixup, ResNet w/o BN with mixup can achieve comparable performance with ResNet with BN on ImageNet. Could FarkasNet further improve the performance in the setting of ResNet w/o BN with mixup and fixup init? This would much more improve the application value of the proposed FarkasNet.

2.	For the results of CIFAR-10 and CIFAR-100, the error bar should be added to make the results more convincing.

3.	Figure 7, the training curves seem weird. Why the training error goes up in some stages?


**Experience Assessment:**

I have published in this field for several years.

**Review Assessment: Checking Correctness Of Derivations And Theory:**

I assessed the sensibility of the derivations and theory.

**Review Assessment: Checking Correctness Of Experiments:**

I carefully checked the experiments.

**Review Assessment: Thoroughness In Paper Reading:**

I read the paper thoroughly.

---

> ### Author Response · Authors · 2019-11-11
> **reply to reviewer #3**
>
> Thank you for the thorough review; we will address your criticisms in order.
>
> 1) When using Farkas layers (FLs) in conjunction with BN, we were not expecting improvements; it was just to complete the experimental methodology. Further, we are not trying to claim that FLs can replace BN, only that it motivates exploration into how the layer weights interact with the data. With deeper networks, performance of FLs is actually highlighted (relative to no BN), though agreed not state-of-the-art.
>
> Due to computational limits, we were unable to perform the full extent of the experiments we would have liked. MixUp requires hyper-parameter tuning which we are not equipped to do. It would be interesting to incorporate FixUp + FLs, however our focus was to show that FixUp does not necessarily solve the problem of training deep networks without any normalization, as their methodology requires input data normalization for CIFAR10, which seems unnecessary.
>
> 2) Agreed; those will be added.
>
> 3) We used the DAWNBench training scheme to efficiently train models on ImageNet. At the “spike” the image-size changes (from small to large).

---

### Decision · Program_Chairs · 2019-12-19

**Decision:**

Reject

**Comment:**

This paper proposes a new normalization scheme that attempts to prevent all units in a ReLU layer from being dead. The experimental results show that this normalization can effectively be used to train deep networks, though not as well as batch normalization. A significant issue is that the paper does not sufficiently establish that their explanation for the success of Farkas layer is valid. For example, do networks usually have layers with only inactive units in practice?